Corrected: Author correction

# Molecular architecture of the multifunctional collagen lysyl hydroxylase and glycosyltransferase LH3

Luigi Scietti[1], Antonella Chiapparino [1], Francesca De Giorgi [1], Marco Fumagalli[2], Lela Khoriauli [3], Solomon Nergadze [3], Shibom Basu[4], Vincent Olieric [4], Lucia Cucca[5], Blerida Banushi[6,8], Antonella Profumo[5], Elena Giulotto[3], Paul Gissen [6,7] & Federico Forneris [1]

Lysyl hydroxylases catalyze hydroxylation of collagen lysines, and sustain essential roles in extracellular matrix (ECM) maturation and remodeling. Malfunctions in these enzymes cause severe connective tissue disorders. Human lysyl hydroxylase 3 (LH3/PLOD3) bears multiple enzymatic activities, as it catalyzes collagen lysine hydroxylation and also their subsequent glycosylation. Our understanding of LH3 functions is currently hampered by lack of molecular structure information. Here, we present high resolution crystal structures of full-length human LH3 in complex with cofactors and donor substrates. The elongated homodimeric LH3 architecture shows two distinct catalytic sites at the N- and C-terminal boundaries of each monomer, separated by an accessory domain. The glycosyltransferase domain displays distinguishing features compared to other known glycosyltransferases. Known disease-related mutations map in close proximity to the catalytic sites. Collectively, our results provide a structural framework characterizing the multiple functions of LH3, and the molecular mechanisms of collagen-related diseases involving human lysyl hydroxylases.

[1] The Armenise-Harvard Laboratory of Structural Biology, Department of Biology and Biotechnology, University of Pavia, Via Ferrata 9/A, 27100 Pavia, Italy. [2] Laboratory of Biochemistry, Department of Biology and Biotechnology, University of Pavia, Via Taramelli 3/B, 27100 Pavia, Italy. [3] Laboratory of Molecular Biology, Department of Biology and Biotechnology, University of Pavia, Via Ferrata 9/A, 27100 Pavia, Italy. [4] Swiss Light Source, Paul Scherrer Institut, Villigen 5232, Switzerland. [5] Laboratory of Analytical Chemistry, Department of Chemistry, University of Pavia, Via Taramelli 12, 27100 Pavia, Italy. [6] MRC Laboratory for Molecular Cell Biology, University College London, London WC1E 6BT, UK. [7] UCL Great Ormond Street Institute of Child Health, 30 Guilford Street, London WC1N 1EH, UK. [8] Present address: Translational Research Institute, The University of Queensland Diamantina Institute, Princess Alexandra Hospital, 37 Kent Street, Brisbane, Australia. Correspondence and requests for materials should be addressed to F.F. (email: federico.forneris@unipv.it)

Collagen biosynthesis requires multiple post-translational modifications essential for the generation of mature, triple-helical molecules[1]. Modification of collagen lysines enables subsequent glycosylation and formation of extracellular cross-links, leading to fibrillary or meshwork superstructures[2]. Enzymes belonging to the family of collagen lysyl hydroxylases (LH or PLOD) catalyze lysine hydroxylation of collagens using $Fe^{2+}$, 2-oxoglutarate (2-OG), ascorbate and molecular oxygen[3,4]. In humans, *PLOD* genes encode for three LH enzyme isoforms sharing >60% amino acid sequence identity: LH1, LH2a/b, and LH3, respectively[5]. Mutations in *PLOD* genes that reduce or abolish LH activity are associated with severe connective tissue diseases including Ehlers-Danlos[6] and Bruck syndromes[7,8]. In mouse models, LH3 knock-outs are embrionically lethal[9,10]. Mutations in the *PLOD3* gene also result in impaired collagen glycosylation, secretion, and basement membrane formation, yielding phenotypes resembling osteogenesis imperfecta[11]. Conversely, *PLOD* overexpression and upregulated enzymatic activity have been linked to fibrosis[12], and recently also to hypoxia-induced metastatic spreading of solid tumors with poor prognosis[13–15].

LH3 is considered the evolutionary ancestor of the LH family: this isoform is the only one capable of further processing of hydroxylysines through glycosylation, whereas other isoforms might have lost such capability during evolution[16]. LH3 is therefore a multifunctional enzyme capable of converting collagen lysines into 1,2-glucosylgalactosyl-5-hydroxylysines through three consecutive reactions: hydroxylation of collagen lysines (LH activity), N-linked conjugation of galactose to hydroxylysines (GT activity), and conjugation of glucose to galactosyl-5-hydroxylysines (GGT activity)[17,18]. Biochemical data suggest that these different enzymatic activities are localized in distinct compartments of the enzyme[19], but despite the extensive evidence available, the current knowledge of LH enzymes is far from exhaustive. These enzymes are known to act together with prolyl hydroxylases, respectively introducing hydroxylations of lysine and proline residues on procollagens in the endoplasmic reticulum (ER), prior to the formation of triple-helical assemblies[20]. In line with this, LH enzymes are found as ER-resident proteins albeit they do not possess specific ER-retention sequences[21,22]. Reports suggest that ER retention could be mediated via interaction with distinct ER-resident proteins: LH1 is described to be part of a macromolecular complex with SC65, P3H3 and CYPB[23]; while LH2 forms a complex with HSP47, FKBP65 and BiP[24,25]; LH3 was found colocalizing with collagen galactosyltransferases GLT25D1/2[26]. Multiple reports identify LH3 also in the extracellular space and suggest dedicated trafficking mechanisms for its secretion[27–30]. Abnormalities in LH3 post-Golgi trafficking are associated with devastating developmental diseases with phenotypes characterized by immature collagen accumulation and lack of its secretion, very similar to those observed in case of enzyme malfunctions caused by knock-down or inactivation[9–11,30,31]. Very recently, LH2 secretion has been reported associated with hypoxia-induced *PLOD2* overexpression in metastatic tumor microenvironments[13,15]. Extracellular LHs were reported to be active, suggesting implications for ECM stability and remodeling[27]. These data indicate that although lysine modifications are known to occur in the ER prior to collagen triple helical formation, secreted variants of LH3 and LH2 can modify collagens in different compartments and, possibly, in different folding states[32].

The accumulated knowledge about the precise molecular roles and mechanisms associated with LH enzymes has suffered from the lack of molecular structure models fundamental to shed light on the complexity and the diversity of this important enzyme family. Here, we present the crystal structures of multifunctional full-length human LH3 in complex with various cofactors and donor substrates. The structures reveal a multidomain architecture characterized by two independent catalytic sites devoted to the different enzymatic activities and provide a molecular understanding that has implications for various disease-related mutations found in LH enzymes. Altogether, our results offer new insights into the complex mechanisms of collagen biosynthesis and homeostasis, and provide structural templates for the development of targeted therapies for LH-related diseases and cancer.

## Results

**LH3 has three domains encompassing multiple catalytic sites.** We have generated human stable cell lines for large-scale production of full length, glycosylated human LH3, and established methods for its purification and evaluation of its LH and GT enzymatic activities (Supplementary Fig. 1). ICP-MS analyses indicated that all enzyme preparations contained $Fe^{2+}$ with a 1:1 stoichiometry (see Methods). We observed significant uncoupling (up to 25%) of donor substrate activation, with substrate-independent generation of the succinate or UDP reaction products, respectively. Nevertheless, we could detect significantly increased enzymatic activity in the presence of synthetic peptides (Supplementary Fig. 1C) or gelatin (Supplementary Fig. 1D) as acceptor substrates. We could confirm such reactivity by detection of concentration-dependent binding (Supplementary Fig. 2A) and appearance of post-translationally modified lysine residues on synthetic peptides upon LH3 treatment (Supplementary Fig. 2B). In our peptide binding measurements, all surface plasmon resonance (SPR) profiles were characterized by very fast association and dissociation events and very weak (millimolar) binding affinities (Supplementary Fig. 2A).

Crystal structures of LH3 were determined in complex with various substrates and cofactors at resolutions ranging between 2.1 and 3.0 Å (Supplementary Table 1). Diffraction data were systematically affected by strong anisotropy (Supplementary Fig. 3), thus structure determination required anisotropy correction followed by a combination of experimental phasing with heavy atom and highly redundant native single wavelength anomalous dispersion (SAD), eventually yielding electron density maps of superb quality (Supplementary fig. 4A, B). Residues Asn63 and Asn548 showed extended electron densities protruding from their side chains (Supplementary fig. 4C), indicating the expected N-linked glycosylations. The LH3 monomer encompasses three domains aligned along one direction (Fig. 1a). The first two N-terminal domains show Rossmann-fold architectures reminiscent of glycosyltransferases[33,34], whereas the C-terminal domain is characterized by a double-stranded β-helix (DSBH) fold, highly conserved among the 2-OG, $Fe^{2+}$-dependent dioxygenases[35,36]. Overall, the three-dimensional structure of the LH3 full-length enzyme provides a molecular blueprint to elucidate previous suggestions on the enzyme architecture based on biochemical data: GT and GGT activities localize at the N-terminus of the enzyme, whereas the LH activity is segregated at the LH3 C-terminus[37,38].

**LH3 forms elongated tail-to-tail dimers.** Although the asymmetric units of our LH3 crystal structures contain a single LH3 monomer, previous biochemical studies[20,38] and our size exclusion chromatography coupled to small-angle X-ray scattering (SEC-SAXS) analyses consistently showed 200 kDa dimers in solution (Supplementary Fig. 1A). The crystal packing indeed suggests two homodimeric arrangements with physiologically plausible assemblies (Supplementary Fig. 5). A first, elongated, tail-to-tail quaternary structure shares nearly identical dimer

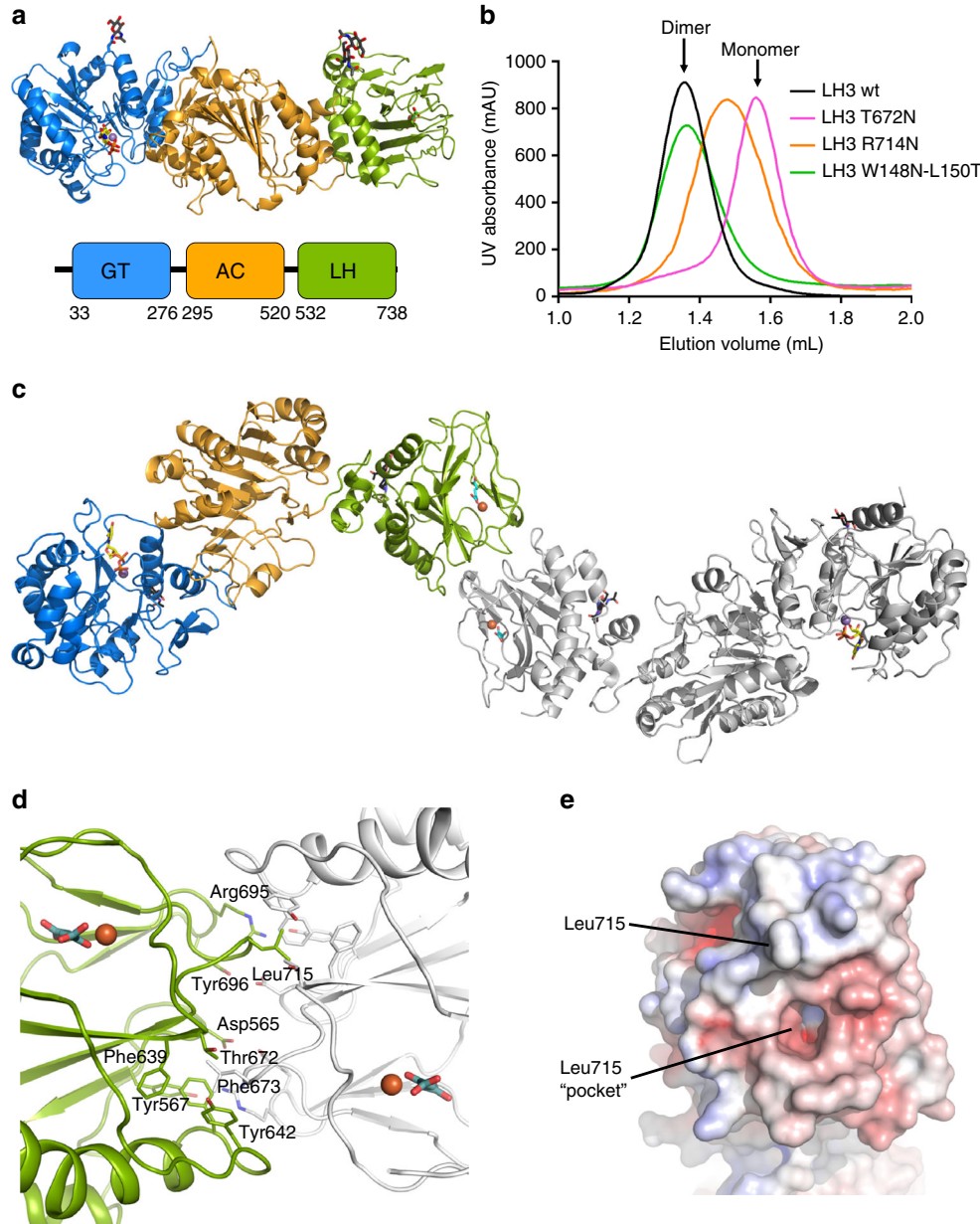

**Fig. 1** Molecular architecture of human LH3. **a** Cartoon representation of the LH3 enzyme, showing its organization with three domains aligned from the N- to the C- terminus. Based on results of functional assays, the first two glycosyltransferase domains have been named GT (catalytic glycosyltransferase, blue) and AC (accessory, orange), respectively. The C-terminal domain hosts the $Fe^{2+}$ and 2-OG cofactors necessary for lysyl hydroxylase activity, and therefore has been named LH (green). Metal ions are shown as spheres, cofactors and glycans as sticks. **b** Introduction of additional glycosylation sites identifies the LH3 dimer interface in solution. Glycosylated mutants R714N and T672N induce disruption of the dimeric assembly, as observed in analytical size exclusion chromatography experiments. Control glycosylated mutant W148N-L150T, located at a crystal contact interface, is a dimer as wild-type LH3. **c** Overview of the LH3 dimer as observed in the crystal structures. The quaternary arrangement highlights an elongated tail-to-tail dimer extending for over 20 nm in one direction, connected through strong electrostatic and hydrophobic interactions near the LH catalytic site. $Fe^{2+}$ and 2-OG in the neighboring catalytic site are shown with spheres and sticks, respectively. For clarity, one LH3 monomer is colored as in **a**, whereas the other is shown in white. **d** The molecular interface connecting two C-terminal LH domains in the LH3 crystal structure is characterized by strong hydrophobic interactions involving Leu715 of one monomer (green) and various aromatic residues shaping a cavity on the opposite monomer (white). This hydrophobic contact is surrounded by electrostatic interactions. For clarity, only the amino acids of one monomer are labeled. **e** Details of the LH3 dimer interface. Shown is the electrostatic potential computed using APBS[70] colored from $-10\ k_bTe_c^{-1}$ (red) to $+10\ k_bTe_c^{-1}$ (blue)

interface with that observed recently in a C-terminal fragment of a viral LH homolog (L230-LH)[39]. This interface interconnects the C-terminal domains of LH3 and exposes individual glycosyl-transferase domains at the two sides of the dimer (Supplementary Fig. 5B). A second, more compact antiparallel conformation is characterized by contacts between the glycosyltransferase

domains and exposes individual lysyl hydroxylase domains at the two sides of the dimer (Supplementary Fig. 5C). Both interfaces are characterized by a large buried surface area and numerous hydrogen bonds and hydrophobic interactions. Previous bio-chemical characterizations based on C-terminal LH3 deletions indicated residues Lys541-Glu547 as essential for dimerization[38].

In our structures, both observed dimeric assemblies fully support this statement, as this region is located in a linking platform connecting the central glycosyltransferase domain with the C-terminal domain (Fig. 1a). Initial attempts using SAXS and computational methods to discriminate between crystallographic contacts and stable dimers in solution (Supplementary Fig. 5–6) were not conclusive. A recent report suggested that in both homologous viral L230-LH domain and in human LH2, replacement of the fully conserved, surface-exposed C-terminal residue Leu715 (LH3 numbering, located in the middle of the "elongated" dimer interface found in the LH3 crystal structure) with a charged Asp could disrupt the enzyme's dimeric assembly and generate inactive, monomeric species in solution[39]. We took advantage of this information and introduced the corresponding L715D mutation in LH3; we also opted for generating a second mutant, bearing a positive Arg side chain replacing Leu715. Surprisingly, the L715D mutant was comparable to wild-type LH3 in activity assays and in analytical size exclusion chromatography experiments, while we observed slightly increased retention volumes and abolished LH activity for the L715R mutant (Supplementary Fig. 7). Similarly, removal of $Fe^{2+}$ using chelating agents and acidification, successfully exploited to destabilize both viral LH and human LH2 dimer interfaces[39] did not seem to affect LH3 stability (Supplementary Fig. 7C). We therefore decided to introduce more pronounced steric hindrance at the observed crystallographic dimer interfaces, through mutations carrying additional glycosylation sites. Mutants T672N and R714N, both adding a glycosylation near the C-terminal dimer interface as verified by SDS-PAGE and enzymatic deglycosylation experiments (Supplementary Fig. 7A, B, Supplementary Fig. 17), shifted the size exclusion retention volumes towards monomeric species (Fig. 1b) and abolished LH activity (Supplementary Fig. 7D). On the contrary, the LH3 mutant W148N-L150T, introducing an additional glycosylation at the "compact" N-terminal interface did not affect the dimeric quaternary structure of LH3 nor the LH enzymatic activity (Fig. 1b, Supplementary Fig. 7D). We therefore concluded that the physiological LH3 dimeric assembly corresponds to the elongated, tail-to-tail arrangement shown in Fig. 1c. This interface is characterized by two-fold symmetric interactions involving a rather limited set of aminoacid side chains engaged in electrostatic contacts, plus a deep hydrophobic cavity shaped by Phe673, Phe639, Tyr642 and Thr672 hosting the side chain of Leu715 from the opposite monomer (Fig. 1d, e).

**Structural insights into LH3 glycosyltransferase activity.** The N-terminal LH3 glycosyltransferase domain partially shares tertiary structure topology with divalent metal ion-dependent class-A glycosyltransferase folds (GT-A)[33], but with distinguishing structural features as expected given the very low sequence identity conservation, lower than 12%. Indeed, although numerous three-dimensional structures of GT-A glycosyltransferases are available in the protein data bank, superpositions with even the closest structural homolog yielded root mean square deviations (r.m.s.d.) higher than 3 Å (Supplementary Fig. 8A). Notably, this LH3 domain lacks a highly conserved α-β hairpin at its N-terminus near residue Gly70, and includes other structural elements surprisingly well conserved at the primary sequence level within the LH enzyme family (Supplementary Fig. 9), but distinct from other glycosyltransferases. In particular, the conformations of four loops differ from GT-A structures and shape the substrate-binding face of the N-terminal domain of LH3 (Supplementary Fig. 8A). Among these, a very flexible surface loop comprising residues Gly72 to Gly87 is not visible in the electron density of ligand-free LH3 structures. This loop contains several residues highly conserved among LH isoforms (Supplementary

Fig. 9), which are not found in other glycosyltransferases. A cavity characterized by aspartate residues 112 and 115 and His253 shapes the metal ion binding site. Co-crystallizations with $Mn^{2+}$ resulted in appearance of strong electron density proximate to these residues for the metal ion and two coordinating water molecules, without observable conformational changes compared to ligand-free LH3 (Supplementary Fig. 10A). Co-crystallizations with $Mn^{2+}$ and donor substrates UDP-galactose or UDP-glucose yielded additional clear electron densities for UDP, but not for the glycan moieties (Fig. 2a). We observed weak electron density near the UDP pyrophosphate group partially compatible with glycan donor substrates, but we refrain from modeling anything inside this weak density, likely representative of multiple conformations simultaneously trapped in the substrate binding cavity (Supplementary Fig. 10B). We could not detect significant differences when comparing co-crystal structures obtained using UDP-Gal or UDP-Glc donor substrates. Nonetheless, binding of these donor substrates induced dramatic conformational changes in the enzyme's catalytic site, with full stabilization of the flexible Gly72-Gly87 loop in a "closed" conformation (Fig. 2b). The UDP pyrophosphate group is stabilized by interactions with $Mn^{2+}$ and hydrogen bonding with Lys259 and backbone nitrogen of Gly256; both residues are positioned in a uniquely shaped α-helix located at the C-terminus of the domain. The hydroxyl groups of the ribose form a network of hydrogen bonds with backbone atoms of Ser113 and Tyr114. The uracil moiety is sandwiched through π-π stacking interactions between Trp75 and Tyr114, and is stabilized by hydrogen bonding with Thr46 (Fig. 2a, b, Supplementary Fig. 10). Of note, these two residues highlight an unprecedented arrangement of UDP binding residues in glycosyltransferases: Trp75 belongs to the distinctive LH3 flexible loop covering residues Gly72-Gly87, that becomes fully stabilized upon substrate binding (Fig. 2b); Tyr114 is part of a non-canonical DxxD motif (Supplementary fig. 9), where Asp112, and Asp115 are responsible for $Mn^{2+}$ coordination. Structurally related glycosyltransferases also often bear tyrosine residues stacking with the UDP moiety, but these residues are located far in sequence from the canonical DxD motif[40,41] responsible for metal ion coordination. Site-directed mutagenesis on Trp75 or Tyr114 into alanine residues yielded folded, but almost completely inactive LH3 variants (Fig. 2c, Supplementary fig. 11). Binding data using SPR on synthetic collagen peptides showed very limited differences between wild-type and mutant LH3 (Supplementary Fig. 11D). Together, these results highlight the distinguishing roles of Trp75 and Tyr114 in donor substrate binding and stabilization.

Two tryptophan residues Trp145 and Trp148 proximate to the UDP-binding cavity consistently change their side chain conformations in structures with bound donor substrates (Fig. 2b). In UDP-glycan-bound structures, residue Trp145 adopts a conformation that can easily accommodate the sugar moiety during catalysis (Fig. 2b). In the absence of donor substrates this residue partially obstructs the cavity, thus acting as a gating residue to host the UDP-glycan (Fig. 2b). Trp148 localizes on the LH3 surface, in a region distant from the glycosyltransferase catalytic site located at the interface between two crystallographically related LH3 molecules. Interestingly, both Trp145 and Trp148 residues are located in one of the loops that are not conserved neither in glycosyltransferases, nor in other LH enzymes (Supplementary Fig. 9): this region may therefore have a role in recognition and binding of donor and/or acceptor substrates, highlighting the unusual mechanisms of LH3 GT/GGT activity. Of note, mutagenesis studies aimed at characterizing LH3 GT/GGT activities showed that residues in this loop are indeed critical for LH3-mediated collagen glycosylation[19] (Supplementary Table 2). The neighboring region

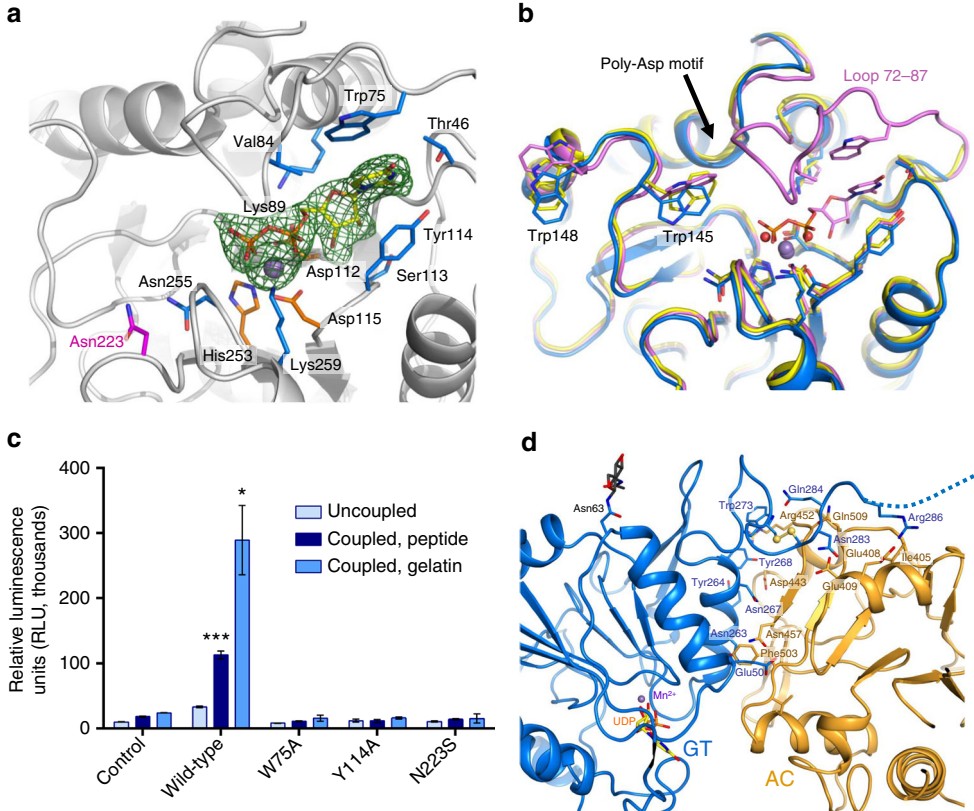

**Fig. 2** Insights into LH3 glycosyltransferase activity. **a** Co-crystallizations with $Mn^{2+}$ and donor substrates revealed clear electron density ($2F_o$-$F_c$ omit electron density maps, green mesh, contour level 1.2 σ) for the metal ion and for UDP in the catalytic site of the N-terminal GT domain. Residues involved in coordination of the metal ion are shown with orange sticks, while residues interacting and stabilizing UDP binding are shown in blue. Residue Asn223, found mutated into a Ser and causing pathogenic phenotypes similar to osteogenesis imperfecta, is shown with magenta sticks. **b** Binding of donor substrates induces conformational changes in the GT domain. Shown is a superposition of ligand-free (yellow), $Mn^{2+}$-bound (blue), and $Mn^{2+}$-donor substrate-bound (violet) structures of LH3 GT domain. Only the UDP-Gal-bound structure is shown; the conformation observed in UDP-Glc-bound structure is identical. Flexible loop 72–87 becomes well defined only in donor substrate-bound structures; in these structures, residues Trp145 and Trp148 adopt different conformations. **c** Luminescence-based assays for the evaluation of LH3 GT enzymatic activity show that mutants W75A, Y114A, and N223S are inactive. Control experiments were performed without adding enzyme. Error bars represent standard deviations from average of triplicate independent experiments. Statistical evaluations based on pair sample comparisons between uncoupled and coupled assay values using Student's *t*-test. *P-value <0.05; ***P-value <0.001. **d** Details of the interface between the N-terminal (GT, blue) and the central (AC, orange) LH3 glycosyltransferase domain. Residues found at the interface are shown as sticks (side chain view only). The disulfide bond found in the linker region between the two domains is shown with yellow spheres on the sulfur atoms

comprising residues 187–191 is characterized by a non-conserved poly-Asp repeat (Supplementary Fig. 10B), and mutations on Asp190 and Asp191 were reported to abolish the glycosyltransferase activity[19] (Supplementary Table 2). In UDP-bound structures, this short loop forms contacts with the stabilized Gly72-Gly87 loop directly involved in UDP interaction (Fig. 2b). Although Asp190 and Asp191 are not directly involved in binding of enzyme cofactors and substrates, both residues point towards the active site near Trp145, possibly playing roles as nucleophiles during the glycosyltransferase reaction, or supporting solvent-bridged interactions with the glycan co-substrates.

Interestingly, the pathogenic LH3 mutation N223S, responsible for an LH3-dependent developmental connective tissue disorder with phenotype resembling osteogenesis imperfecta[11] localizes in close proximity to the identified LH3 GT/GGT catalytic site (Fig. 2a). This mutation was reported to introduce a new N-linked glycosylation site on residue Asn221, resulting in strongly reduced LH3 GT/GGT activity (Supplementary Table 2). We produced this enzyme variant obtaining a folded, dimeric enzyme comparable with wild type LH3 (Supplementary Fig. 11A, B). However, due to pronounced instability and high propensity to

degradation for this mutant, we could not unambiguously confirm that this disease-linked mutant bears an additional glycosylation (Supplementary Fig. 11A). LH3 N223S showed severely reduced lysyl hydroxylase and fully abolished glycosyltransferase activities (Fig. 2c, Supplementary Fig. 11C). As we did not observe changes in LH3 oligomeric assembly in the presence of this pathogenic mutation (Supplementary Fig. 11B), we concluded that the lack of enzymatic activity caused by this variant is likely due to the alterations in enzyme stability, possibly introduced by the novel glycosylation on Asn221, which would interfere with recognition of acceptor substrate molecules.

The C-terminal part of the first glycosyltransferase domain of LH3 incorporates other uniquely structured regions: a long, non-conserved beta hairpin constituted by residues Val229-Ala244 points towards the domain face opposite to the GT/GGT catalytic site. This segment is stabilized by numerous hydrophobic contacts including Phe233, Trp273 and with a non-conserved α-helix formed by residues Pro257-Glu276, which is sandwiched between the first and the second domain of LH3 (Fig. 2d). Residues linking these two domains extend to a solvent-exposed region of the enzyme comprising the Cys279-Cys282 disulfide

bridge and a flexible loop extending from Asp285 to Gly292, and not visible in the electron density. Antibodies targeting these residues were reported to reduce GT activity[17], suggesting a role for this surface-exposed region in modulating accessibility of acceptor substrates to the identified glycosyltransferase catalytic site (Supplementary Table 2).

In the central LH3 domain, two previously suggested candidate metal ion binding sites are present, identified by a DxD (Asp392, Ala393, Asp394) and a DxDxD (Asp486, Thr487, Asp488, Pro489, Asp490) motif, respectively[18,19] (Supplementary Fig. 12). Co-crystallization experiments with metal ions and glycosyl-transferase substrates and products did not highlight any appreciable differences within this domain compared to ligand-free structures (Supplementary Fig 13A). On the contrary, superposition of the two glycosyltransferase domains of LH3 showed remarkable differences, emphasized by the unique features found in the first domain (Supplementary Fig. 8, Supplementary Fig. 13B). Although the overall fold of the second domain shares higher similarity than the first with known GT-A type glycosyltransferases (Supplementary Fig. 13C, D), extensive mutagenesis experiments did not allow clear identification of residues implicated in metal ion or substrate binding for LH3 GT catalytic activity[18,19] (Supplementary Table 2). Our structures highlight an unusual conformation for the Val304-Phe310 loop: this region overlaps with the donor substrate binding site observed in structural homologs, strongly interfering with donor substrate binding (Supplementary Fig. 13E). Furthermore, recombinant N-terminal LH3 constructs devoid of this domain

were still capable of glycosylating collagen peptides similar to the wild-type enzyme[19] (Supplementary Table 2). Collectively, these data indicate that despite the overall conservation of the glycosyltransferase fold, this domain may have lost its enzymatic capabilities during evolution. It may therefore constitute a non-catalytic accessory element within the LH3 architecture, possibly involved in collagen substrate recognition or interactions with LH binding partners. Thus, we named the LH3 N-terminal domain catalytic glycosyltransferase (GT), and the central domain accessory (AC).

**Structural insights into LH3 lysyl hydroxylase activity.** The C-terminal domain of LH3 hosts the lysyl hydroxylase catalytic site of the enzyme. This domain shows the typical DSBH fold, characterized by two β-sheets with antiparallel β-strands flanked by three α-helices (Fig. 3a). This fold is preceded by two additional short helices covering residues Thr523 to Asp554, serving as a buffer platform between the AC and the LH domain (Fig. 1a). Residues 590–610 constitute a flexible loop capping the LH catalytic site. Superposition with identified structural homologs highlighted the high conservation of the overall fold and the consistent presence of highly flexible residues near the Fe$^{2+}$, 2-OG binding site (Supplementary Fig. 14A). The recently determined structure of a viral L230-LH domain[39] shows very high overall similarity (r.m.s.d. = 1.3 Å), with the exception of surface-exposed residues Ser550-Ile558, likely due to the non-conserved N-linked glycosylation at residue Asn548 (Supplementary Fig. 14B); also the structure of the viral LH fragment shows a

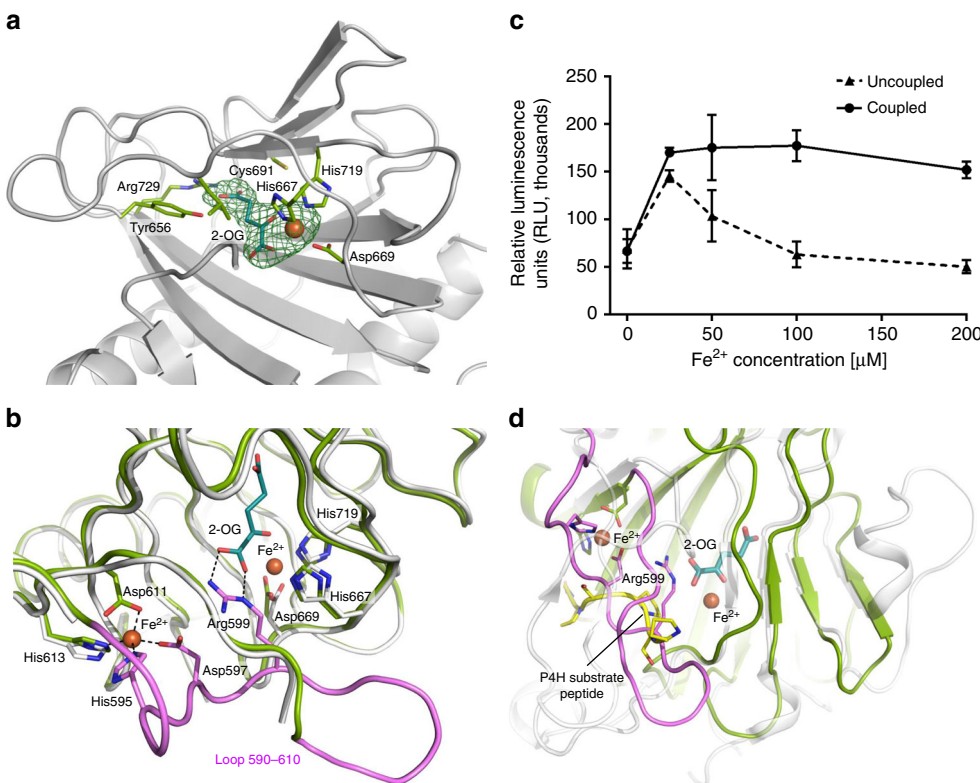

**Fig. 3** Insights into LH3 lysyl hydroxylase activity. **a** All our LH3 structures consistently show clear electron density for bound Fe$^{2+}$ and 2-OG in the LH catalytic cavity (2F$_o$-F$_c$ omit electron density maps, green mesh, contour level 1.2 σ). Residues involved in interactions with Fe$^{2+}$ and 2-OG are shown with green sticks. **b** Co-crystallizations with Fe$^{2+}$ allow identification of a second metal ion bound near the LH catalytic site that stabilizes the flexible capping loop 590–610 (shown in pink). **c** Evaluation of collagen substrate coupled and uncoupled LH3 LH activities as a function of Fe$^{2+}$ concentration. Error bars represent standard deviations from average of triplicate independent experiments. **d** In Fe$^{2+}$ co-crystal structures, the conformation observed for residue Arg599 mimics the collagen lysine substrate in front of the 2-OG cofactor, yielding a non-productive ternary complex. Shown is the superposition of this conformation found in the LH3 LH domain (green) with an homologous algal prolyl-4-hydroxylase (white) in complex with a short poly-PS peptide (yellow)[43]. The LH3 loop 590–610 is shown in pink

completely flexible capping loop. All our LH3 structures systematically showed $Fe^{2+}$ and 2-OG in the electron density near the core of this domain (Fig. 3a), confirming that these cofactors are tightly bound to the enzyme and providing a possible explanation for the observed substrate uncoupled LH enzymatic activity (Supplementary Fig. 1C, D). Given the high structural similarity of our LH3 structure with the LH domain of the viral variant, we were surprised not to find the 2-OG cofactor in this structure. We interpreted this difference as due to usage of a prokaryotic expression system for the recombinant production of the viral homolog[39]. In human LH3, the $Fe^{2+}$ ion is stabilized by interactions with residue His719 and with residues His667 and Asp669, which constitute part of a HxD motif fully conserved in the $Fe^{2+}$, 2-OG dioxygenase enzyme family (Supplementary Fig. 15). Mutations in these residues[17], including a recently identified pathogenic variant of LH2 causing Bruck syndrome[8], were found to completely abolish LH activity (Supplementary Table 2). Pathogenic LH3 mutations causing premature C-terminal enzyme truncations lacking His719 were found incapable of hydroxylating collagen lysines, although retaining GT/GGT activities[11] (Supplementary Table 2). Residues Tyr656, Cys691 and Arg729 are fully conserved among LH isoforms and delimit the pocket hosting the 2-OG cofactor (Supplementary Fig. 14C). In particular, the side chain of free Cys691 contributes to the positioning of the carboxyl moiety of the 2-OG cofactor in the direction of Arg729, forming a salt bridge with the guanidinium group of this residue. In addition, biochemical mutations of Arg729 in homologous LH1 (Arg719 in LH3) were reported to cause complete loss of LH activity due to decreased binding affinity for 2-OG[42]; these data are also in agreement with a recent report analyzing these mutations in the viral L230-LH homolog[39] (Supplementary Table 2).

**Excess $Fe^{2+}$ induces a state showing substrate mimicry**. By increasing the $Fe^{2+}$ concentration in crystallization experiments, we serendipitously found that this metal ion contributes to the overall enzyme stabilization, systematically enhancing the quality and the resolution of X-ray diffraction data. Analysis of electron density maps allowed the identification of a second $Fe^{2+}$ in the LH domain, coordinated by residues His595, Asp597, Asp611 and His613 (Fig. 3b). Metal ion coordination stabilizes the flexible capping loop 590–610 into a conformation that completely plugs the LH catalytic site, in proximity to the dimer interface (Supplementary Fig. 16A). We could observe a very similar arrangement, although slightly more flexible, by replacing $Fe^{2+}$ with $Mn^{2+}$ in crystallization experiments (Supplementary Fig. 16B). Early reports indicated that LH enzymes may be inhibited by high concentrations of metal ions[20]. We probed LH3 enzymatic activity in the presence of increasing concentrations of $Fe^{2+}$, and found that the enzymatic activity peaks at 25 μM $Fe^{2+}$ concentration. Higher metal ion concentrations yield significant reduction of LH uncoupling, but do not seem to affect the enzymatic reactivity in the presence of synthetic peptide substrates (Fig. 3c). In the metal-ion stabilized conformations we also found that residue Arg599 forms a salt bridge with the 2-OG co-substrate, yielding a conformation that may mimic the collagen lysine substrate. The positioning of Arg599 indeed superimposes to that of a collagen proline residue as observed in a homologous prolyl hydroxylase structure in complex with a short peptide[43] (Fig. 3d).

## Discussion

Despite being known for over 40 years[44], several molecular aspects underlying collagen lysyl hydroxylases function are still obscure. Our crystal structures of full-length human LH3

rationalize the accumulated biochemical knowledge, offering a template to better understand the molecular mechanisms of LH3-dependent collagen hydroxylation and glycosylation.

The two different enzymatic activities are segregated in two distinct catalytic sites 80 Å apart within the same enzyme molecule. The overall dimeric arrangement, experimentally validated through site-directed mutagenesis, shows an elongated tail-to-tail quaternary assembly (Fig. 1c), characterized by a relatively small hydrophobic contact platform surrounded by electrostatic contacts that stabilize the dimer interface near the LH catalytic site (Fig. 1c–e) and closely resemble those recently reported for homologous human LH2 and viral L230[39] (Supplementary Fig. 14B). As previously suggested, dimerization is essential for LH activity, whereas disruption of physiological dimers does not significantly perturb the N-terminal glycosyltransferase activities of LH3[38,39].

We identified two glycosyltransferase domains at the N-terminus of the enzyme characterized by Rossmann fold-like domain architecture and partial similarity with known GT-A glycosyltransferases. However, only the first domain is active: the GT domain (residues 33–277) is the solely responsible for glycosyltransferase activities (both GT and GGT), albeit showing unique features in multiple regions of its fold strongly divergent from known glycosyltransferases (Supplementary Fig. 8). Sequence alignments comparing human LH isoforms (Supplementary Fig. 9) and analysis of donor substrate-free and bound structures do not provide a direct explanation for the lack of GT/GGT activities in homologous LH1 and LH2. Nearly all residues surrounding the UDP moiety and shaping the LH3 glycosyltransferase catalytic site are conserved, including those involved in the unprecedented mode of UDP-substrate stabilization characterized by dual π-π stacking with Trp75 and Tyr114. Nevertheless, sequence conservation at the rim of the catalytic site is much lower (Supplementary Fig. 9), suggesting that lack of collagen substrate recognition may be the reason for the absence of glycosyltransferase activity in homologous LH isoforms.

The adjacent AC domain does not appear to be directly involved in enzyme function. Surprisingly, this domain shares the highest structural similarities with known glycosyltransferases, and yet it may have lost its function during evolution through mutations that introduced steric hindrance in the donor and acceptor substrate binding regions (Supplementary Fig. 13E). Despite the lack of direct evidence of substrate recognition capabilities, the elongated quaternary structure of the LH3 enzyme allows to speculate about possible long-range collagen substrate recognition mechanisms involving this domain. Alternatively, it may serve as a docking module for interactions with binding partners, such as the proposed collagen glycosyltransferases GLT25D1/2[26] or chaperones like FKBP65[24,25].

In the C-terminal LH domain both $Fe^{2+}$ and 2-OG are natively retained and consistently found in the electron densities of all our structures (Fig. 3a). Enzymatic assays show that LH activity is present without the need for $Fe^{2+}$ supplementation, demonstrating that the metal ion remains tightly bound into the active site during catalysis, and supporting recent reports also indicating essential roles for the metal ion in folding maintenance of LH domain[39]. Concentrations higher than 25 μM affect the LH3 uncoupled catalytic activity, but do not seem to affect substrate processing. Crystals grown with excess $Fe^{2+}$ or $Mn^{2+}$ show stabilization of a substrate gating loop, with formation of a self-inactivated ternary complex between $Fe^{2+}$, 2-OG and the side chain of Arg599 mimicking the collagen lysine substrate and obstructing the enzyme's active site (Fig. 3b, c). Despite gating loops have been described for prolyl-hydroxylases[43,45,46], the presence of a metal ion-induced gating seems a prerogative of LH enzymes. Given the full conservation in the LH enzyme family of

amino acid residues involved in coordination of the second $Fe^{2+}$ (Supplementary Fig. 15), we postulate that a general mechanism for metal ion-dependent regulation of LH activity may rely on generation of a self-inactivated resting state with interlocking of the 591–610 gating loop in front of the catalytic site. Remarkably, mutation of LH2 conserved homologous Arg599 into a histidine, as well as other mutations localized on this gating loop, perturb the enzymatic activity and cause Bruck syndrome[7] (Supplementary Table 2). This corroborates the proposed role of the 590–610 gating loop in modulating substrate accessibility to the LH active site, and in particular the role of conserved Arg599 as non-productive substrate mimicry.

The LH3 structures allow understanding the molecular phenotypes associated to most of the pathogenic mutations involving LH enzymes (Fig. 4). The LH3 N223S variant causing connective tissue disorders similar to osteogenesis imperfecta was reported to introduce a novel N-linked glycosylation near the cavity hosting the glycosyltransferase donor substrates. Our experiments show that this alteration strongly affects the enzyme's stability, causing degradation, and completely abolishes GT and GGT activities. Pathogenic mutations of LH1 responsible for the kyphoscoliotic variant of the Ehlers-Danlos syndrome were reported to reduce, but not abolish, mechanisms of collagen recognition and catalysis (Supplementary Table 2). This is fully consistent with their localization on the enzyme's surface, distant from the identified catalytic sites, with the exception of pathogenic mutation H706R, which maps to a conserved residue in close proximity to the homodimer interface. Conversely, LH2 mutations causing severe Bruck syndrome are located within the LH active site, either proximate to the 2-OG binding site, directly involved in $Fe^{2+}$ coordination, or in the self-inactivated capping loop, also including the lysine mimicry residue Arg599 (Supplementary Table 2). Given the recent reports describing implications of human LH2 and its upregulated enzymatic activity in metastatic spreading of numerous solid tumors[13], our full-length structures of self-inhibited human LH3 constitute a valid template for structure-based drug discovery campaigns aiming at blocking unwanted collagen lysine hydroxylation in tumor microenvironments.

In conclusion, the crystal structures of full-length human LH3 in complex with various cofactors and donor substrates provide a molecular understanding of the biochemical knowledge underlying the multiple functions of this enzyme. Our data shed light on the unique molecular architecture of the LH3 glycosyltransferase domains, and allow understanding of the molecular

bases of multiple genetic diseases involving LH3 and homologous human lysyl hydroxylases.

## Methods

**Chemicals.** All chemicals were purchased from Sigma-Aldrich unless otherwise specified.

**DNA constructs.** Human LH3 gene (GenBank accession number BC011674.2) was obtained from Source Bioscience. Oligonucleotides containing in-frame 5′-BamHI and 3′-NotI were designed and used to sub-clone the LH3 sequence devoid of the N-terminal signal peptide into the pUPE.106.08 expression vector, kindly provided by U-protein Express, BV (U-PE, The Netherlands) and into the pPuro-DHFR, containing the *Streptomyces alboniger* puromycin resistance gene (isolated from the pPUR plasmid, Clontech) and the mouse dihydrofolate reductase cDNA[47]. Both expression vectors bear N-terminal signal peptide followed by a N-terminal 6xHis-tag and a recognition site for Tobacco Etch Virus (TEV) protease prior to an in-frame BamHI restriction site, as well as an in-frame stop codon after the NotI restriction site. LH3 mutants were generated using the Phusion Site Directed Mutagenesis Kit (ThermoFisher Scientific) following manufacturer's instructions. The entire plasmid was amplified using phosphorylated primers. For all mutants the forward primer introduced the mutation of interest (Supplementary table 4). All expression plasmids were checked by Sanger sequencing prior to usage.

**Recombinant LH3 expression from stable HeLa cell lines.** The pPuro-DHFR-LH3 construct was transfected into human cervical carcinoma cells (HeLa S3, provided by ATCC and further selected for high transfection efficiency by Dr.F. Peverali, Consiglio Nazionale delle Ricerche, Pavia) using the Lipofectamine LTX reagent (Invitrogen). Cells were not authenticated and not tested for myco-plasma contamination. Stably transfected clones, isolated with 1 mg mL$^{-1}$ pur-omycin (Invivogen), were subjected to step-wise selection with increasing methotrexate concentrations to select for cells containing multiple copies of the plasmid. Cells were routinely cultured at 37 °C in 5% CO2 in high-glucose DMEM supplemented with 10% foetal calf serum (Biowest), 1× non-essential amino acids, 2 mM L-glutamine and 1× penicillin-streptomycin. Clones expressing high yields of PLOD3 were identified by SDS-PAGE analysis after imidazole elution from small-scale immobilized metal ion affinity purification using Nickel sepharose beads (GE Healthcare).

**Recombinant LH3 expression from transient HEK293 cells.** Recombinant tag-ged LH3 mutants were produced using suspension cultures of HEK293F (Invi-trogen) cells. Cells were not authenticated and not tested for mycoplasma contamination. Cells were transfected at cell densities of 1 million mL$^{-1}$ using 3 µg of polyethyleneimine (PEI; Polysciences, Germany) for 1 µg of pUPE.106.08-LH3 plasmid DNA per mL of cells. Cultures were supplemented with 0.6% Primatone RL 4 h after transfection. The cell medium containing secreted LH3 was collected 6 days after transfection by centrifugation at $1000 \times g$ for 15 min.

**Purification of LH3 enzymes.** The LH3-containing medium from either HeLa or HEK293 cell cultures was filtered through a syringe 0.2 mm filter. The pH and ionic strength of the filtrated medium were adjusted using a concentrated buffer stock to reach a final concentration of 25 mM 4-(2-hydroxyethyl)−1-piper-azineethanesulfonic acid (HEPES)/NaOH, 500 mM NaCl, 30 mM imidazole, pH

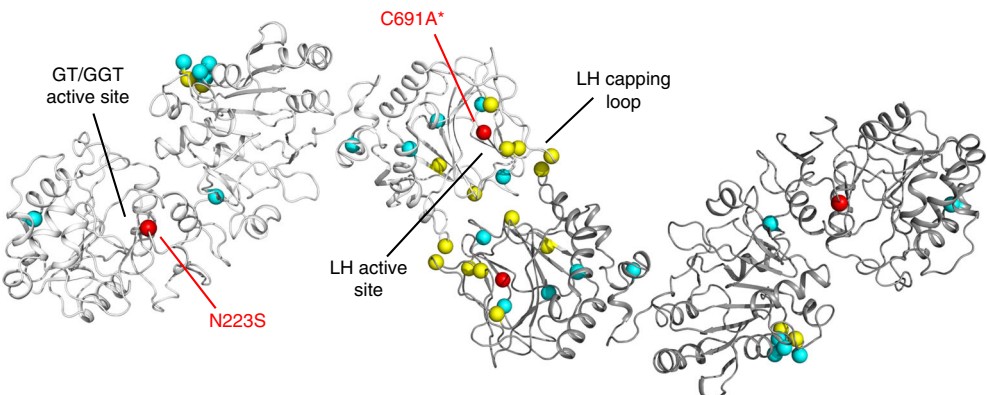

**Fig. 4** Mapping of disease-related mutations identified in LH enzymes on the LH3 crystal structure. The positions of pathogenic mutations causing Ehlers-Danlos type VI syndrome in LH1 (cyan), Bruck syndrome type II in LH2 (yellow), and connective tissue diseases sharing phenotype features with osteogenesis imperfecta (red) in LH3 are shown as spheres on the LH3 dimer structure. Mutations N223S and C691A*, found on LH3, are labeled in red in one of the two monomers. A complete list of mutations reported in the literature and their significance based on evaluation of their mapping on the LH3 crystal structure is shown in Supplementary Table 2

8.0. LH3 was purified using affinity and size-exclusion chromatography on Äkta systems (GE Healthcare). The filtered supernatant was first loaded onto a 20 mL His-Prep FF column (GE Healthcare) and eluted using 250 mM imidazole. The eluate was then loaded onto a 5 mL HiTrap desalting FF column (GE Healthcare) equilibrated in 25 mM HEPES, NaOH, 500 mM NaCl, pH 8.0. The N-terminal histidine-tag was cleaved using overnight TEV protease digestion at 4 °C followed by affinity-based removal of TEV protease and His-tag using a 5 mL HisTrap FF (GE Healthcare). The protein was concentrated to 5 mg mL$^{-1}$ using 30,000 MWCO Vivaspin Turbo centrifugal filters (Sartorius), then loaded onto a Superdex 200 10/300 GL (preparative scale) or onto a Superdex 200 5/150 GL (analytical scale) columns (GE Healthcare) equilibrated with 25 mM HEPES/NaOH, 200 mM NaCl, pH 8.0. LH3-containing fractions as assessed from SDS-PAGE analysis were pooled, concentrated and stored at −80 °C until further usage.

**LH3 deglycosylation.** Wild-type LH3 and mutants T672N, L714N, and W148N-L150T were subject to deglycosylation to validate the introduction of an additional glycosylation site through mutagenesis. 20 μL protein at 0.15 mg mL$^{-1}$ were first incubated with 1X Glycoprotein Denaturing Buffer (New England BioLabs) and denatured at 95 °C for10 min. 1X glycobuffer 2 (New England BioLabs), 1% NP-40, and 0.2 μL (100 Units) of PNGase F (New England BioLabs) were added to the reaction mix, which was further incubated for 2 h at 37 °C. PNGase F-treated and untreated samples were then analyzed using western blotting with rabbit anti-LH3 antibody (Proteintech 11027–1-AP) in a 1:1000 ratio followed by anti-rabbit HRP-conjugate antibody (Sigma-Aldrich A0545) in a 1:3000 ratio.

**ICP-MS measurements.** To measure the number of tightly bound divalent metal ions 3 mg of LH3 were diluted in 5 mL 25 mM HEPES, 500 mM NaCl, pH 8.0. Three aliquots of 1 ml each were kept overnight with 0.5 mL 65% ultra-pure HNO$_3$ and 0.1 mL 30% w/w H$_2$O$_2$, then diluted to 5 mL with Milli-Q water and analyzed by ICP-MS. The measurements of Fe and Mn were performed on a Perkin Elmer Mod ELAN DRC-e instrument, following the standard procedures suggested by the manufacturer. Quantitative determinations were obtained by the external standard calibration with five standards (0, 5, 10, 50, 100, and 300 μg L$^{-1}$) daily prepared in the same buffer used for samples preparation, at the same dilution and HNO$_3$ concentration. Only Fe was quantified in each replicate solution, with standard deviation of the mean value of 8%, obtaining a molar ratio Fe/LH3 = 1, while Mn was present as impurity in the blank and in the samples.

**Crystallization of LH3.** LH3 spherulites were found in nanoliter-dispensed droplets (0.1 μL protein at 4 mg mL$^{-1}$ + 0.1 μL reservoirs) using commercial crystallization screens in sitting vapor diffusion drop plates. These spherulites were initially optimized by mixing 0.5 μL of protein concentrated at 3.5 mg mL$^{-1}$ and 0.5 μL of reservoir solution composed of 600 mM sodium formate, 12% polyglutamic-acid (PGA-LM, Molecular Dimensions), 100 mM HEPES/NaOH, pH 7.8. These crystals diffracted to a maximum of 5 Å. Crystal quality was improved through sequential runs of macro-seeding of LH3 crystals using the same crystallization mixture with slight variations in protein concentration. Co-crystallization experiments were performed by setting up the same seeding conditions, and supplementing the protein solution with mixtures of 500 μM FeCl$_2$, 500 μM MnCl$_2$, 1 mM UDP-galactose, 1 mM UDP-glucose (Supplementary Table 1). Crystals were cryo-protected with the mother liquor supplemented with 20% glycerol, harvested using MicroMounts Loops (Mitegen), flash-cooled and stored in liquid nitrogen prior to data acquisition. Heavy atom derivatives were prepared by soaking the LH3 crystals in mother liquor conditions containing 1 mM K$_2$HgBr$_4$. Crystals were incubated with the heavy atom solution for at least 5 h at 4 °C prior to cryo protection, harvesting and flash-cooling in liquid nitrogen.

**Diffraction data collection and structure refinement.** Diffraction data from LH3 crystals were collected at various beamlines of the European Synchrotron Radiation Facility, Grenoble, France and at the Swiss Light Source, Villigen, Switzerland (details in Supplementary Table 1). Single wavelength Anomalous Dispersion (SAD) experiments at the Hg edge were performed at the ESRF ID30B beamline, whereas high multiplicity long-wavelength native SAD data (57 data sets of 360° from 6 different crystals) were collected at the SLS X06DA beamline as described elsewhere[48]. The data, which showed strong anisotropy (Supplementary Fig. 3), were processed with autoPROC[49] including STARANISO[50]. Statistics are summarized in Suppl. table 1. Hg heavy atom sites were identified using SHELXC/D[51] with the HKL2MAP GUI[52]. Experimental phasing with the Hg SAD data using SHELXE[51] and SHARP[53] yielded a partial model. Completion of the model could only be achieved by combining the latter with high multiplicity native SAD data using the CRANK2 pipeline[54] followed by iterations of automatic and manual model building using BUCCANEER[55] and COOT[56]. Subsequent LH3 structures were determined using the initial LH3 structural model in molecular replacement runs with PHASER[57]. Final 3D models were generated using iterations of automatic refinement using PHENIX[58] alternated with manual adjustments using COOT[56]. Validation of structure quality was carried out with Molprobity[59], the RCSB PDB Validation Server[60], and PDB-CARE[61]. Final refinement statistics are listed in Suppl. Table 1. Structural figures were generated using PyMol[62].

**SAXS data collection and analysis.** Solution scattering data were collected at ESRF BM29 using a sec$^{-1}$ frame rate on Pilatus 1 M detector located at a fixed distance of 2.87 m from the sample, allowing a global $q$ range of 0.03–4.5 nm with a wavelength of 0.01 nm. SEC-SAXS experiments were carried out using Nexera High Pressure Liquid/Chromatography (HPLC; Shimadzu) system connected online to SAXS sample capillary[63]. For these experiments, 50 μL of LH3 concentrated at 4 mg mL$^{-1}$ were injected into a Superdex 200 PC 3.2/300 Increase column (GE Healthcare), pre-equilibrated with 25 mM HEPES/NaOH, 200 mM NaCl, pH 8.0. For offline batch sample analysis, 50 μL of LH3 at concentrations ranging from 0.5 to 9 mg mL$^{-1}$ were injected using the dedicated automatic sample changer available at the BM29 beamline[64]. For SEC-SAXS data, frames corresponding to LH3 protein peak were identified, blank subtracted and averaged using CHROMIXS[65], whereas batch concentration series were analyzed using PRIMUS[66]. Radii of gyration ($Rg$), molar mass estimates and distance distribution functions $P$ ($r$) were computed using the ATSAS package[67] in PRIMUS[66]. Comparison of experimental SAXS data and 3D models from crystal structures was performed using CRYSOL[68]. A summary of SAXS data collection and analysis results is shown in Suppl. Table 3.

**Determination of LH activity using mass spectrometry.** Synthetic collagen peptides were purchased from China peptides. Peptides tested were ARGIK-GIRGFS, GIKGIKGIKGIK, and IKGIKGIKG sequences. LH3 5 μM was incubated with 500 μM FeCl$_2$, 1 mM 2-OG, 2 mM ascorbate and 1 mM peptide substrate. Reactions were allowed to proceed for 1 h at 37 °C. In total 20 μL of each sample were previously acidified by addition of 1 μL of formic acid (FA) and then analyzed on an LC–MS system (Thermo Finnigan, USA) consisting of a thermostated column oven Surveyor autosampler controlled at 25 °C; a quaternary gradient Surveyor MS pump equipped with an UV/vis detector and an Ion Trap (LCQ Advantage Max) mass spectrometer with electrospray ionization ion source controlled by Xcalibur software 2.0.7. Peptides were separated by RP-HPLC on a Jupiter (Phenomenex, USA) C$_{18}$ column (150 × 2 mm, 4 μm, 90 Å particle size) using a linear gradient (2–60% solvent B in 60 min) in which solvent A consisted of 0.1% aqueous FA and solvent B of acetonitrile (CAN) containing 0.1% FA. Flow-rate was 0.2 mL/min. Mass spectra were generated in positive ion mode under constant instrumental conditions: source voltage 5.0 kV, capillary voltage 46 V, sheath gas flow 20 (arbitrary units), auxiliary gas flow 10 (arbitrary units), sweep gas flow 1 (arbitrary units), capillary temperature 200 °C, tube lens voltage −105 V. Spectra analyses were performed using Xcalibur software 2.0.7.

**Biochemical evaluation of LH activity.** Reaction mixtures (5 μL total volume) containing wild-type or mutant LH3 samples at 0.2 mg mL$^{-1}$ were prepared by sequentially adding 0–1 mM peptide substrate or 4 mg mL$^{-1}$ gelatin in water, (solubilized through heating denaturation at 95 °C for 10 min), 500 μM ascorbate, 100 μM 2-OG, and variable concentrations of FeCl$_2$ (0–200 μM), and let incubate for 1 h at 37 °C. Reactions were stopped by heating samples at 95 °C for 2 min prior to transfer into Proxiplate white 384-well plates (Perkin-Elmer), then 5 μL of the Succinate-Glo reagent I (Promega) were added and let incubate 1 h at 25 °C, after that 10 μL of the Succinate-Glo reagent II (Promega) were added and let incubate 10 min at 25 °C. The plates were then transferred into a GloMax plate reader (Promega) configured according to manufacturer's instructions for luminescence detection. All experiments were performed in triplicates. Control experiments were performed using identical conditions by selectively removing LH3, 2-OG or peptide substrates. Data were analyzed and plotted using the GraphPad Prism 7 software[69].

**Biochemical evaluation of GT and GGT enzymatic activities.** Reaction mixtures (5 μL total volume) containing wild-type or mutant LH3 samples at 0.2 mg mL$^{-1}$ were prepared by sequentially adding 0–1 mM peptide substrate, 500 μM ascorbate, 100 μM 2-OG, 50 μM FeCl$_2$, and let incubate for 1 h at 37 °C. Reactions mixtures were then supplemented with 50 μM MnCl$_2$, and 50 μM UDP-Gal or UDP-Glc, and let incubate for 1 h at 37 °C. Experiments using gelatin as substrate were performed by sequentially adding 4 mg mL$^{-1}$ gelatin in water, (solubilized through heating denaturation at 95 °C for 10 min), 50 μM MnCl$_2$, and 100 μM UDP-Gal or UDP-Glc to the LH3 samples at 0.2 mg mL$^{-1}$, and let incubate for 1 h at 37 °C. All reactions were stopped by heating at 95 °C for 2 min, prior to transfer into Proxiplate white 384-well plates (Perkin-Elmer), then 5 μL of the UDP-Glo luminescence detection reagent (Promega) were added and let incubate 1 h at 25 °C. Detection was carried out as described for the LH enzymatic activity. All experiments were performed in triplicates. Control experiments were performed using identical conditions by selectively removing LH3, donor or acceptor substrates. Data were analyzed and plotted using the GraphPad Prism 7 software[69].

**Surface-plasmon resonance.** Wild-type and mutant LH3 preparations were immobilized onto a carboxymethylated dextran (CM5) sensor chip (GE Healthcare) using a mixed solution of 200 mM 1-ethyl-3-(3-dimethylaminopropyl)carbodiimide hydrochloride (EDC) in 50 mM N-hydroxysuccinimide (NHS) in a Biacore T200 SPR instrument (GE Healthcare). Excess reactive groups were blocked with 1 M ethanolamine. Efficient immobilization of LH3 samples was judged based on the SPR signal collected. For each of the tested samples, 3000 RU

were reached. A control flow cell 1 was pre-activated and blocked using the same protocol as above but without protein samples, and used as reference cell during measurements. Collagen peptides were dissolved in running buffer (PBS-P 0.01%) and injected as two-fold dilution concentration series of eight points each using a flow of 5 µl min$^{-1}$. Two replicates of each concentration were injected. Data analysis was performed using the Biacore T200 evaluation software (GE Healthcare) using a 1:1 steady-state affinity model.

**Data availability**. Coordinates and structure factors have been deposited in the Protein Data Bank (PDB) with accession codes 6FXK, 6FXM, 6FXR, 6FXT, 6FXX, 6FXY. SEC-SAXS experimental data and ab-initio model have been deposited in Small Angle Scattering Biological Data Bank (SASBDB) with accession code SASDDW4. Other data are available from the corresponding author upon reasonable request.

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

## Acknowledgements

We thank scientists at ARDIS S.R.L. Pavia for technical support during the SPR experiments. We thank the European Synchrotron Radiation Facility (ESRF) and the Swiss Light Source (SLS) for the provision of synchrotron radiation facilities and beamline scientists of the SLS, ESRF and the European Molecular Biology Laboratory for assistance. In particular, we would like to thank M. Brennich (EMBL Grenoble) and P. Pernot (ESRF) for support with SAXS data collection and analysis. We thank M. Miao for support in crystallization experiments and Prof. A. Mattevi for useful discussions. This work was supported by the Giovanni Armenise-Harvard Career Development Award, the "Programma Rita Levi-Montalcini" from the Italian Ministry of University and Research (MIUR), Cariplo Foundation Grant "COME TRUE" (id. 2015-0768), a "My First AIRC Grant" grant (Grant id. 20075) from the Italian Association for Cancer Research (AIRC), and by the Italian Ministry of Education, University and Research (MIUR): Dipartimenti di Eccellenza Program (2018–2022)—Dept. of Biology and Biotechnology "L. Spallanzani", University of Pavia. For X-ray diffraction experiments we were supported by the European Community's Seventh Framework Programme (FP7/2007-2013) under BioStruct-X (Grant agreements 7551 and 10205). A.C. is supported by a Marie Curie Individual Fellowship from the Horizon 2020 EU Program (Grant agreement no. 745934 – COTETHERS). P.G. is supported by the NIHR HS&DR Programme (14/21/45) and supported by the NIHR GOSH BRC. The views expressed are those of the authors and not necessarily those of the NHS, the NIHR or the Department of Health.

## Author contributions

L.S., A.C., B.B., L.K., F.F. performed LH3 cloning and expression trials. L.K., S.N., and E.G. designed and implemented the stable HeLa cell lines for LH3 production. L.S. and F. F. purified and crystallized LH3, and solved its structure with help from S.B. and V.O. L. S. and F.F. carried out structural refinement. A.C. generated and purified LH3 mutants using HEK293 cells. L.S. and F.D.G. performed biochemical assays. M.F. performed mass spectrometry experiments. L.C. and A.P. performed ICP-MS metal ion identification experiments. L.S. performed surface-plasmon resonance measurements. F.F. designed the study with help from B.B. and P.G. L.S., A.C., F.D.G., and F.F. analyzed the data and prepared figures. L.S., A.C. and F.F. wrote the paper, with contributions from all authors.

## Additional information

**Competing interests:** The authors declare no competing interests.

