## [Peer Review File · Nature Communications]

Reviewers' comments:

Reviewer #1 (Remarks to the Author):

The manuscript by Scietti et al. reports the long-awaited crystal structure of a full-length human lysyl hydroxylase, LH3. This bifunctional enzyme is essential for the biosynthesis of functional collagen, as illustrated by the severe disorders that result from mutated LH3. The structure shows that the two enzymatic activities reside at opposite ends of an extended three-domain structure. The lysine hydroxylase domain is similar to other Fe²⁺/2-oxoglutarate-dependent hydroxylases, but the Glc/Gal-transferase domain has many unique and interesting features. The central domain also has a glycosyltransferase fold, but is not catalytically active (a fact that is neatly explained by the structure). Overall, the study by Scietti et al. represents a major advance in understanding LH3, but the presentation of results needs serious attention before publication can be recommended.

1. LH3 is a dimer in solution and the crystal packing suggests two alternative dimers. Scietti et al. made a considerable effort to define the physiological dimer, but their results are inconclusive. SAXS experiments are compatible with either dimer, and no mutation is reported that disrupts the dimer (the pathogenic N223S mutation abolishes catalytic activity but not dimerisation). The matter appears to be settled by a recent structure of a viral LH (Guo et al., Nat. Commun., 2018) that shows the same dimer interface as the "elongated" in Scietti's study. Guo et al. reported that mutation of key leucine disrupts the dimer. Rather than dwelling on their ambiguous experiments, Scietti et al. should mutate the equivalent leucine in LH3 (Leu715, prominently illustrated in Fig. 1E) and determine the effect of this mutation on dimerisation. Reference to the second ("compact") dimer in the Discussion should be removed unless the authors present experimental evidence that this dimer is more than a crystal lattice artefact. If the authors wish to retain Supplementary Figures 5 and 6, a number of points would need to be addressed: a) Were glycans included in the SAXS modelling shown in Fig. S4C? b) Does the smeared high-MW band for N223S in Fig. S5A disappear upon PNGase F treatment? c) What is the point of the "inset" in Fig. S5C? How are these data different from the ones shown in Fig. 2C? d) What exactly are the "synthetic peptides" used in the SPR experiments in Fig. S5D?

2. I could not find a single figure that shows the location of Asn223, even though the N223S mutant is included in Fig. 2C and discussed at length in the manuscript.

3. In Fig. 1, it would be more logical to show the dimer as panel C (currently D) and the dimer interface as panel D (currently C). As mentioned in point 1, the role of Leu715 in dimerisation should be confirmed experimentally.

4. Are the Fe²⁺ concentrations in Fig. 3C correct? I assume Fe²⁺ concentrations are micromolar and not millimolar (there is also reference to 25 mM Fe²⁺ at the bottom of page 8). Is there any evidence that the "gating mechanism" by transition metal ions is physiological?

5. The MS data in Fig. S1D are not very convincing. LH3 treatment increases the peak at +16 Da slightly, but the reaction appears to be far from complete. Much better data can be found in the literature, e.g. Fig. 3b of Mantri et al., J. Mol. Biol., 2010.

6. The diffraction data are reported to be strongly anisotropic. Could this statement be quantified, e.g. by quoting CC1/2 and I/ (I) values along h, k and l?

7. The interactions in the dimer interface are not "palindromic" (page 5) but "two-fold symmetric".

8. The "consensus" in the alignment in Fig. S7 is no such thing; it indicates identities and similarities.

9. Regarding Fig. 8B, does the density shown correspond to UDP-Gal or UDP-Glc? It would be helpful to indicate the likely conformation of the donor sugar. I assume this information can be inferred from other glycosyltransferase structures.

Reviewer #2 (Remarks to the Author):

Major breakthrough in structural analysis of vertebrate collagen hydroxylases! Impressive success in crystallization of full-length human collagen lysyl-hydroxylase 3! The structure is of fundamental importance for understanding LH-related diseases, as well as fibrosis and cancer progression. Foundation for intelligent design of drugs. Detailed analysis and comprehensive supporting data on enzyme structure and activity. Would be of great interest for not only collagen related research community. Well-written and thoughtful. Will provoke further research on collagen post-translational modifications. Highly recommend for publication.

Minor points:

1. LH1 and LH2 complexation with other ER-resident proteins is not mentioned in Introduction.
2. Are there indications for LH3 complexation with other ER-resident proteins? Since there is a signal peptide and no ER-retention signal I don't see necessity to point for "non-conventional trafficking mechanisms for its secretion".
3. "All enzyme preparations were found to contain Fe²⁺ with a 1:1 stoichiometry." How was it found? Nothing in Methods or Results.
4. Suppl. Fig. 5A. "The presence of smeared high molecular weight bands indicating the additional glycosylation of mutant N223S." Not convincing as there are multiple bands present for this particular mutant preparation possibly indicating impurities.
5. Suppl. Fig. 5D. Y115A should be Y114A
6. Suppl. Fig. 8B. Side chain labels are missing for residues mentioned in the text.
7. Suppl. Fig. 13B. What are pink balls?
8. "Nanoliter-dispensed droplets (0.1 nL protein at 4 mg/mL + 0.1 nL reservoirs) using commercial crystallization screens in sitting vapor diffusion drop plates." Personally I never heard of such systems. Was it really 0.1nL, not 0.1uL? What kind of system was it? Please provide more details of such fantastic system.

POINT-BY-POINT RESPONSE TO REVIEWER'S COMMENTS

Reviewer #1

The manuscript by Scietti et al. reports the long-awaited crystal structure of a full-length human lysyl hydroxylase, LH3. This bifunctional enzyme is essential for the biosynthesis of functional collagen, as illustrated by the severe disorders that result from mutated LH3. The structure shows that the two enzymatic activities reside at opposite ends of an extended three-domain structure. The lysine hydroxylase domain is similar to other Fe²⁺/2-oxoglutarate-dependent hydroxylases, but the Glc/Gal-transferase domain has many unique and interesting features. The central domain also has a glycosyltransferase fold, but is not catalytically active (a fact that is neatly explained by the structure). Overall, the study by Scietti et al. represents a major advance in understanding LH3, but the presentation of results needs serious attention before publication can be recommended.

1. LH3 is a dimer in solution and the crystal packing suggests two alternative dimers. Scietti et al. made a considerable effort to define the physiological dimer, but their results are inconclusive. SAXS experiments are compatible with either dimer, and no mutation is reported that disrupts the dimer (the pathogenic N223S mutation abolishes catalytic activity but not dimerisation). The matter appears to be settled by a recent structure of a viral LH (Guo et al., Nat. Commun., 2018) that shows the same dimer interface as the "elongated" in Scietti's study. Guo et al. reported that mutation of key leucine disrupts the dimer. Rather than dwelling on their ambiguous experiments, Scietti et al. should mutate the equivalent leucine in LH3 (Leu715, prominently illustrated in Fig. 1E) and determine the effect of this mutation on dimerisation. Reference to the second ("compact") dimer in the Discussion should be removed unless the authors present experimental evidence that this dimer is more than a crystal lattice artefact.

We would like to thank the reviewer for this suggestion. In the revised version of our manuscript, we have addressed this matter more carefully by adding a number of experiments to validate the LH3 dimeric assembly in solution. We indeed generated the Leu715Asp mutant suggested by others, but we intriguingly found that this mutation was not sufficient to disrupt the LH3 dimer interface. We therefore generated additional mutations bearing glycosylation sites at dimer interfaces and we eventually ruled out the second "compact" dimer as crystal contact. Text in the **results and discussion sections** (line 139-157 "*We took advantage of [...] from the opposite monomer (Figure 1D-E)*" and line 314-318 "*The overall dimeric arrangement [...] for homologous human LH2 and viral L230 (Supplementary Fig. 14B)*"), as well as **Figure 1B have been changed accordingly**. **Supplementary Fig. 7**, describing a comprehensive overview of the mutants generated, has been added.

If the authors wish to retain Supplementary Figures 5 and 6, a number of points would need to be addressed:

We have **removed the sphere models** previously included in Supplementary Fig. 4B, but we wish to keep the CRYSOLO-based comparison between our crystal structure and the X-ray scattering data, **now shown in Supplementary Figure 6B** of our revised manuscript. We have **changed the text in our revised manuscript** accordingly (lines 134-135 "*Initial attempts using SAXS [...] were not conclusive*").

a) Were glycans included in the SAXS modelling shown in Fig. S4C?

Given the possible high flexibility of glycan antennae, we decided not to model them prior to CRYSOLO analysis. To clarify this point, we **changed the text** describing the content of Supplementary Fig. 6A (previously S6C) as follows: "*Comparison of the crystallographic LH3 dimers with experimental SEC-SAXS data using CRYSOLO shows that none of the crystal-derived assemblies can unambiguously match the state adopted by the protein in solution*".

b) Does the smeared high-MW band for N223S in Fig. S5A disappear upon PNGase F treatment?

We have removed the discussion about the N223S mutation in the context of validation of LH3 dimeric assembly. PNGase F treatment of this mutant, as suggested by the reviewer, did not yield a conclusive outcome. We therefore refrain from using the putative glycosylation introduced by this mutation as supportive of our biochemical observations. We apologize for the confusion generated in the first version of our manuscript on this topic.

c) What is the point of the "inset" in Fig. S5C?

We have **changed the representation styles** for the biochemical data presented in our revised manuscript. We apologize for the confusion created in this figure.

How are these data different from the ones shown in Fig. 2C?

Panel 2C shows the galactosyltransferase (GT) activity measured on the ARGIKGIRGFS mutant for wild-type LH3 and for the mutants, and now includes also measurements on gelatin as substrate. Supplementary Figure 11C (former S5C) instead shows the lysyl hydroxylase (LH) activity of these mutants, on the same peptide, and now includes also evaluation of glucosylgalactosyltransferase activity (GGT) on gelatin substrates. To emphasize differences in biochemical assays for the different enzymatic activities, we introduced a color coding throughout our revised manuscript: black-and-white plots for LH activity; shades of blue for GT activity; shades of green for GGT activity.

d) What exactly are the "synthetic peptides" used in the SPR experiments in Fig. S5D?

Throughout our revised manuscript, we have clarified this point. We have now included **additional supplementary figures** illustrating the SPR binding curves LH3 with the IKGIKGKIKG, GIKGKIKGKIKG, ARGIKGIRGFS peptide substrates for wild-type LH3 and two mutants. We have also **expanded our set of biochemical data** showing complete details of LH enzymatic activity on these three synthetic peptides in Supplementary Fig. 1C, and we have included new measurements of individual GT and GGT enzymatic activities on gelatin (Supplementary Fig. 1D) to demonstrate that our assays can be used to detect both glycosyltransferase reactions.

2. I could not find a single figure that shows the location of Asn223, even though the N223S mutant is included in Fig. 2C and discussed at length in the manuscript.

In the revised version of our manuscript, we have **modified Figure 2A** with more clear labels showing the position of the N223S pathogenic mutation in the LH3 GT catalytic site. We also decided to move the supplementary figure containing the mapping of pathogenic mutations on LH enzymes to the main text: it is the **new Figure 4**.

3. In Fig. 1, it would be more logical to show the dimer as panel C (currently D) and the dimer interface as panel D (currently C). As mentioned in point 1, the role of Leu715 in dimerisation should be confirmed experimentally.

We thank the reviewer for this suggestion. **We made changes as requested**. Panel C of our revised figure 1 now shows the experimentally confirmed LH3 dimer. Previous panel C has now been moved to panel D in figure 1. We also performed experimental validation of dimer interfaces, as described in Figure 1B and Supplementary Fig. 7 (see also previous reply on this topic).

4. Are the Fe^{2+} concentrations in Fig. 3C correct? I assume Fe^{2+} concentrations are micromolar and not millimolar (there is also reference to 25 mM Fe^{2+} at the bottom of page 8).

We thank the reviewer for noticing these typos. In both cases, the reported concentrations were micromolar and not millimolar. We have **corrected both figure and text** in the revised version (line 301).

Is there any evidence that the "gating mechanism" by transition metal ions is physiological?

This is indeed something we are extremely keen on investigating and we will do in the future. However, we believe that addressing such question with comprehensive data would require efforts that are far beyond revision of the current manuscript. – **no changes made**.

5. The MS data in Fig. S1D are not very convincing. LH3 treatment increases the peak at +16 Da slightly, but the reaction appears to be far from complete. Much better data can be found in the literature, e.g. Fig. 3b of Mantri et al., J. Mol. Biol., 2010.

The reviewer is right, the reactions are far from being complete. However, we would like to point out that we observed this phenomenon systematically, with multiple peptides (as now better shown in the **new supplementary figure 2B**), and we find such observation relevant within the biochemical characterization of LH3 we are presenting. In this respect, the data cited by the reviewer (Mantri et al., 2010) refer to a different enzyme (JMJD6) and different, non-collagen substrates. To assess whether the experimental conditions we used for the assay could have an effect on reaction completion, we also performed a test in the exact same conditions as those reported in Mantri et al., 2010, and observed no differences, except for reduced signal-to-noise ratios due to lower peptide concentrations.

6. The diffraction data are reported to be strongly anisotropic. Could this statement be quantified, e.g. by quoting $CC1/2$ and $I(I)$ values along h , k and l ?

In our revised manuscript, we have **added new Supplementary Fig. 3**, illustrating the anisotropy observed in the diffraction data for all presented datasets.

7. The interactions in the dimer interface are not "palindromic" (page 5) but "two-fold symmetric".

In our revised manuscript, we have **modified the description** of the interactions in the LH3 as suggested by the reviewer (line 155).

8. The "consensus" in the alignment in Fig. S7 is no such thing; it indicates identities and similarities.

We used the word "consensus" as provided and explained in the CLUSTALW FAQ section: <https://www.ebi.ac.uk/Tools/msa/clustalw2/help/faq.html#22>. – We **added a similar explanation in the legend of supplementary figures** containing alignments.

9. Regarding Fig. 8B, does the density shown correspond to UDP-Gal or UDP-Glc? It would be helpful to indicate the likely conformation of the donor sugar. I assume this information can be inferred from other glycosyltransferase structures.

In the **legend of revised Supplementary Fig. 10B** (former S8B), we have specified that "Shown is the $2F_o - F_c$ omit electron density maps (purple mesh, contour level 1.0σ) in the glycan binding site of LH3 co-crystallized in presence of UDP-Gal". We have tried to infer the conformation of the donor sugar by downloading and superposing UDP-Gal and UDP-Glc molecules found in other glycosyltransferase structures, and we have **included a model of a possible conformation for the UDP-Gal in Supplementary Fig. 10B**. However, the electron density found in the LH3 cavity is only partially located in positions compatible with such UDP-Gal conformation, and additional density is present in a position near Trp145, not matching the sugar conformations found in UDP-sugar-bound glycosyltransferases. We therefore believe that at present we should avoid discussing possible conformations of the donor substrate based on the weak electron density found in our structures. We are keen on carrying out such careful investigation in future work. – **no other changes made**.

Reviewer #2

Major breakthrough in structural analysis of vertebrate collagen hydroxylases! Impressive success in crystallization of full-length human collagen lysyl-hydroxylase 3! The structure is of fundamental importance for understanding LH-related diseases, as well as fibrosis and cancer progression. Foundation for intelligent design of drugs. Detailed analysis and comprehensive supporting data on enzyme structure and activity. Would be of great interest for not only collagen related research community. Well-written and thoughtful. Will provoke further research on collagen post-translational modifications. Highly recommend for publication. Minor points:

1. *LH1 and LH2 complexation with other ER-resident proteins is not mentioned in Introduction.*
2. *Are there indications for LH3 complexation with other ER-resident proteins?*

We have **added a sentence in the introduction** (line 67-70) of our revised manuscript addressing the points raised by the reviewer: “*Reports suggest that ER retention [...] galactosyltransferases GLT25D1/2²⁹.*”

Since there is a signal peptide and no ER-retention signal I don't see necessity to point for “non-conventional trafficking mechanisms for its secretion”.

We agree with the reviewer, and therefore we have **changed our sentence** in the revised manuscript into “*dedicated trafficking mechanisms for its secretion*” (line 71).

3. *“All enzyme preparations were found to contain Fe²⁺ with a 1:1 stoichiometry.” How was it found? Nothing in Methods or Results.*

In the revised version of our manuscript, we have included **a new paragraph in the methods section** (line 486-495) describing the ICP-MS evaluation of LH3 preparations to assess the content of metal ions and associated stoichiometry, and **changed the sentence in the main text** into a clearer statement: “*ICP-MS analyses indicated that all enzyme preparations contained Fe²⁺ with a 1:1 stoichiometry*” (line 94-95). In view of these changes, we **added as co-authors of our manuscript** Dr. L. Cucca and Prof. A. Profumo (previously mentioned in the acknowledgements), who performed these experiments and provided the accurate description included in the methods section.

4. *Suppl. Fig. 5A. “The presence of smeared high molecular weight bands indicating the additional glycosylation of mutant N223S.” Not convincing as there are multiple bands present for this particular mutant preparation possibly indicating impurities.*

We agree with the reviewer and we therefore modified the entire analysis associated to the N223S mutation. **Please see also the reply to reviewer 1 on the same topic.**

5. *Suppl. Fig. 5D. Y115A should be Y114A*

Corrected.

6. *Suppl. Fig. 8B. Side chain labels are missing for residues mentioned in the text.*

We thank the reviewer for this observation. In our revised manuscript, we have **modified our reference to main and supplementary figures** in order to maintain the side chain labels of the various figures as provided.

7. *Suppl. Fig. 13B. What are pink balls?*

The description of the pink spheres is present in Supplementary Fig. 16B (formerly S13B) legend: “*...Although with increased flexibility (as highlighted by lack of density and associated molecular model for amino acids 592-596 and 603-605, model boundaries shown with pink spheres), the Mn²⁺-bound structure shows a very similar arrangement of this capping loop.*” – **No changes made.**

8. *“Nanoliter-dispensed droplets (0.1 nL protein at 4 mg/mL + 0.1 nL reservoirs) using commercial crystallization screens in sitting vapor diffusion drop plates.” Personally I never heard of such systems. Was it really 0.1nL, not 0.1uL? What kind of system was it? Please provide more details of such fantastic system.*

We thank the reviewer for noticing this typo, which has been **corrected in the revised version** “...*Nanoliter-dispensed droplets (0.1 μ L protein at 4 mg mL⁻¹ + 0.1 μ L reservoirs)*...” (line 498). Indeed, sub-nanoliter volume dispensers for protein crystallization would be fantastic. Unfortunately, they do not exist.

LIST OF CHANGES MADE BASED ON EDITORIAL MANUSCRIPT CHECKLIST

- We have changed the title of our manuscript to remove punctuation
- We have shortened the abstract to make it no more than 150 words and compliant with editorial checklist description
- We have changed mathematics by expressing unit dimensions using negative integers
- We have changed subheadings in “results” and “methods” sections to make them shorter than 60 characters and without punctuation
- We have changed format of reference display materials as requested
- We have added statement about data availability
- We have reduced the number of references in the main text to 70

REVIEWERS' COMMENTS:

Reviewer #1 (Remarks to the Author):

The authors have done a good job in revising their manuscript.

The dissociation into monomers as a result of the T672N mutation is convincing evidence that the elongated dimer is the physiological one. Please note that this mutant should be labelled LH3 T672N in Figure 1B, and not LH4 T672N.

The expanded Biacore and MS data in Supplementary Figure 2B are much more convincing than the original submission.

The presentation of results is much clearer throughout.